# Expectation Alignment: Handling Reward Misspecification in the Presence of Expectation Mismatch

**Malek Mechergui, Sarath Sreedharan**
Colorado State University
Fort Collins, 80523
{Malek.Mechergui, Sarath.Sreedharan}@colostate.edu

## Abstract

Detecting and handling misspecified objectives, such as reward functions, has been widely recognized as one of the central challenges within the domain of Artificial Intelligence (AI) safety research. However, even with the recognition of the importance of this problem, we are unaware of any works that attempt to provide a clear definition for what constitutes (a) misspecified objectives and (b) successfully resolving such misspecifications. In this work, we use the theory of mind, i.e., the human user's beliefs about the AI agent, as a basis to develop a formal explanatory framework, called Expectation Alignment (*EAL*), to understand the objective misspecification and its causes. Our *EAL* framework not only acts as an explanatory framework for existing works but also provides us with concrete insights into the limitations of existing methods to handle reward misspecification and novel solution strategies. We use these insights to propose a new interactive algorithm that uses the specified reward to infer potential user expectations about the system behavior. We show how one can efficiently implement this algorithm by mapping the inference problem into linear programs. We evaluate our method on a set of standard Markov Decision Process (MDP) benchmarks.

## 1 Introduction

Given the accelerating pace of advancement within AI, creating agents that can detect and handle incorrectly specified objectives has become an evermore pressing problem [Dafoe et al., 2020]. This has resulted in the development of several methods for addressing, among other forms of objective functions, misspecified rewards (cf. Hadfield-Menell et al. [2017]). Unfortunately, these works operate under an implicit definition of what constitutes an incorrectly specified reward function. The few works that have tried to formalize how an agent could satisfy human objectives do so by avoiding the objective specification step altogether [Hadfield-Menell et al., 2016].

Our primary objective with this paper is to start with an explicit formalization of what constitutes a misspecified reward function and the potential reasons why the user may have provided one in the first place. However, our motivation for providing such a formalization goes beyond just the pedagogical. We see that such a formalization provides us with multiple insights of practical importance.

Our formulation will follow the intuition set by recent work (cf. [Abel et al., 2021]) that looks at reward functions as a means of specifying a task as opposed to being an end to itself. As such, the user starts with a target outcome or behavior or a set of outcomes or behaviors [1] that they want the agent to achieve. The reward function proposed by the user is one whose maximization, they believe,

---

[1] In this paper, we will focus on cases where expected/desired outcomes correspond to the agent achieving certain states. Section 3 provides a formal definition.

38th Conference on Neural Information Processing Systems (NeurIPS 2024).

will result in a policy that will achieve the desired outcome. Figure 1 shows how users derive their reward specifications. As presented, their reward function will be informed by their beliefs about the agent and its capabilities (i.e., the user's theory of mind [Apperly and Butterfill, 2009] about the agent) and their ability to reason about how the agent will act given a specific reward function.

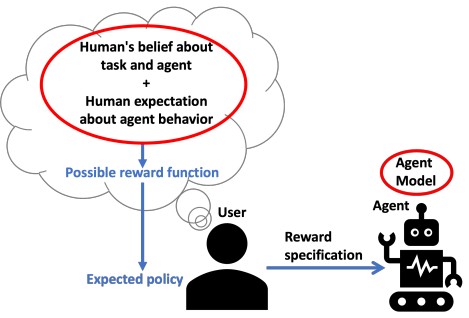

Figure 1: A diagrammatic overview of how specifying a reward function plays a role in whether or not their expectations are met.

A given reward function is said to be misspecified when the policy identified by the agent for the reward function doesn't satisfy the user's expectations. As discussed, the reasons for such misspecification could include users' incorrect beliefs about the agent and their limited inferential capabilities [Sreedharan, 2022]. To see how such misspecification might occur, consider a simple planning problem that can be captured by a Markov Decision Process (MDP) with only three states. Consider a problem where an agent can only perform two possible actions. These correspond to pressing two different switches. Each switch takes the agent to one of the two possible end states where the agent can no longer act. One corresponds to a safe end state, while the other cor-

responds to an unsafe one. Let's consider a human supervisor who wants the agent to go to the safe end state and, as such, identifies a reward function that they believe will result in such a state. If the supervisor is confused about which button leads to which state, they may end up coming up with a reward function that will result in the exact opposite outcome. Please note that the problem here isn't that the agent can't achieve the desired outcome but rather that optimizing for the specified reward will not result in the desired outcome. In fact, given the human's confusion about the buttons, no reward function exists that could lead the agent to the desired outcome, which the human will agree is a correct reward function for the task.

We will formalize this intuition about reward specification and, by extension, that of misspecification under the more general framework of *Expectation Alignment* (*EAL*). *EAL* framework will allow us to capture scenarios, like the one discussed above, where the human cannot come up with a reward that satisfies the expectation in both their and the agent's models. This invalidates all approaches that try to identify a 'true' human reward function, which is then passed onto the agent. Secondly, we will see how we can use *EAL* to develop novel algorithms for handling reward misspecification that explicitly leverages potential causes of misalignment to come up with more effective queries to identify the human's original intent. To summarize, the contributions of the paper are:

- We introduce a framework for formalizing and understanding reward misspecification problems – namely, *Expectation Alignment*.

- We develop a novel query-based algorithm to solve a specific instance of *EAL* problems.

- We empirically demonstrate how the method compares against baseline methods for handling reward uncertainty in benchmark domains.

In the related work section (Section 5), we will also show how existing works on handling objective misspecification relate to our proposed framework. Since it will leverage the specifics of our proposed framework, we will look at the related work after defining our basic framework.

## 2   Background

We will primarily focus on problems that can be expressed as a Markov Decision Process or an MDP [Puterman, 2014]. An MDP is usually represented by a tuple of the form $\mathcal{M} = \langle S, A, T, R, \gamma, s_0 \rangle$[2]. Under this notation scheme, $S$ captures the set of states, $A$ the set of actions, $T : S \times A \times S \to [0, 1]$ is the transition function such that $T(s, a, s')$ captures the probability of an action $a$ causing the state to transition from $s$ to $s'$, $R : S \times A \to \mathbb{R}$ is the reward function, $\gamma \in [0, 1)$ the discount factor and finally $s_0 \in S$ is the initial state (the agent is expected to start from $s_0$). Since we are focusing on

---

[2]We can additionally use control constraints to limit the actions possible from each state, as in the example in the introduction. However, we will leave it out of the formulation to support a more concise formulation.

problems related to reward specification, we will separate out all non-reward components of an MDP model, refer to it as the domain, and denote it as $\mathcal{D} = \langle S, A, T, \gamma, s_0 \rangle$.

The objective of solving an MDP is to find a policy (a function of the form $\pi : S \rightarrow A$) that maximizes the expected sum of discounted rewards (captured by a value function $V : S \rightarrow \mathbb{R}$). A policy is considered optimal (optimal policies will be denoted with the superscript '$*$', for example, $\pi^*$) if it results in the highest value (referred to as the optimal value $V^*$). It is worth noting that an optimal policy is not unique, and for a given task, there may be multiple optimal policies with the same value. We will denote the set of optimal policies associated with a model $\mathcal{M}$ as $\Pi^*_{\mathcal{M}}$.

In this paper, we will be heavily utilizing the notion of discounted occupancy frequency or just occupancy frequency, $x^\pi : S \times A \rightarrow [0, 1]$ [Poupart, 2005], associated with a policy $\pi$. There are multiple ways to interpret occupancy frequency, but one of the most common ones is to think of them as the steady state probability associated with a policy $\pi$. In particular, the occupation frequency $x^\pi(s, a)$ captures the frequency with which an action $a$ would be executed in a state $s$ under the policy $\pi$. Sometimes, we will also need to identify the frequency with which a policy $\pi$ would visit a state $s$. We can obtain this frequency by summing over the values of $x^\pi(s, a)$ for all actions, i.e., $x^\pi(s) = \sum_a x^\pi(s, a)$. It is possible to reformulate the value obtained under a policy in terms of its occupancy frequency and rewards. One can also identify the optimal value and, thereby, the optimal policy by solving the following linear program (LP) expressed using occupancy frequency variables:

$$\max_x \sum_{s,a} x(s, a) \times r(s, a)$$
$$\text{s.t.} \quad \forall s \in S, \sum_a x(s, a) = \delta(s, s_0) + \gamma \times \sum_{s',a'} x(s', a') \times T(s', a', s) \tag{1}$$

Here, $x$ is the set of variables that captures the occupancy frequency possible under a policy, and $\delta(s, s_0)$ is an indicator function that returns 1 when $s = s_0$ and zero otherwise.

In this paper, we will be looking at settings where the human's (i.e. the user who is coming up with the reward function) understanding of the task may differ from that of the robot [3]. As such, the information contained within the domain used by the robot might differ from the beliefs of the human. We will use the notation $\mathcal{D}^R = \langle S, A^R, T^R, \gamma^R, s_0 \rangle$ to denote the domain used by the robot, while $\mathcal{D}^H = \langle S, A^H, T^H, \gamma^H, s_0 \rangle$ is a representation of the human beliefs, i.e., the theory of mind they ascribe to the robot. Note that under this notation scheme, we assume that the two models share the same state space and starting state. This was purely done to simplify the discussion. Our basic formulation and the specific instantiation hold as long as a surjective mapping exists from robot to human states. This would usually be the case where the human model is, in fact, some form of abstraction of the true robot model.

## 3 Expectation Alignment Framework

In this section, we will first develop a framework to understand how humans go from expectations to reward specifications, which we will use to define reward misspecification. With the basic model in place, we can effectively invert it to develop methods to address such misspecification.

To build such a forward model, we need a formal method to represent behavioral expectations in the context of MDPs. While several promising mathematical formalisms could be adopted to represent a human's behavioral expectations, we will focus on using the notion of occupancy frequency. There are multiple reasons for this choice. For one, this is a natural generalization of the notions of reachability. Psychological evidence supports that people use the notion of goals in decision-making [Simon, 1977]. While goals relate to the idea of reaching desirable end states, the notion of occupancy frequency allows us to extend it to intermediate ones. It even allows us to express probabilistic notions of reachability. Secondly, occupancy frequencies present a very general notion of problems that can be expressed as MDPs. As discussed in Section 2, the value of any given policy can be represented in terms of their occupancy frequency [Poupart, 2005]. Finally, the popular use cases in the space of reward misspecification can be captured using occupancy frequency (See Section 5).

So, we start by representing the set of human expectations as $\mathbb{E}^H$, which provides a set of states and any preferences placed on reachability for that state. More formally,

---

[3]We use the term robot to denote our AI agent as means of convenience. Nothing in this framework requires the agent to be physically embodied.

**Definition 1.** *Given human's understanding of the task domain $\mathcal{D}^H = \langle S, A^H, T^H, s_0, \gamma^H \rangle$, the **human expectation set** is denoted as $\mathbb{E}^H$, where each element $e$ in the set is given by the tuple of the form $e = \langle S_e, \mathcal{O}, k \rangle$, where $S_e \subseteq S$, $\mathcal{O} \in \{<, >, \leq, \geq, =\}$ is a relational operator and $k \in [0, 1]$. $\mathcal{O}$ and $k$ places limit on the cumulative occupancy frequency over the set of states $S_e$.*

For cases where the human wants the robot to completely avoid a particular state $s$, there would be a corresponding expectation element $\langle \{s\}, =, 0 \rangle$; on the other hand, if the human wants the robot to visit certain states, then there will be expectation elements that try to enforce high occupancy frequencies for those states. In the simplest settings, humans might convert such expectations to rewards by setting low or even negative rewards to the states that need to be avoided and high rewards to the states they want the robot to visit. One question that is worth considering is why humans don't just communicate this expectation set. Unfortunately, communicating the entire set may be relatively inefficient compared to coming up with a reward function (for example, one might be able to compactly represent the reward function as a piece of code, while this set may not). This could also be the case if the number of states is very high or even infinite, but even in these settings, the reward function could still be specified in terms of features. Secondly, even if they choose to specify desirable or undesirable states, the expectation set may contain states that the human incorrectly thinks are impossible (for example, the robot jumping onto the table) and may choose to ignore them. This is also related to the problem of unspecified side-effects [Zhang et al., 2018].

It is worth noting that Definition 1 merely provides a mathematical formulation that is general enough to capture human expectations. We don't expect people to maintain an explicit set of numeric constraints on occupancy frequencies in their minds. Regardless of what form their actual expectation takes, as long as it can be expressed as some ordering on how the states should be visited, it can be captured using sets of the form provided in the above definition. One could, in theory, also capture non-Markovian expectations using such formalisms. For example, there may be cases where the user might want the robot to visit a set of states in a specific sequence. We can capture such cases by creating augmented states that can track whether or not the robot visited the states in sequence. This method also allows Markovian reward functions to capture such considerations (cf. [Abel et al., 2021]), which is unsurprising since the expressivity of occupancy frequency parallels that of Markovian reward functions.

We will claim that a policy satisfies an expectation element for a domain if the corresponding occupancy frequency relation holds for the policy in that domain, or more formally:

**Definition 2.** *A given policy $\pi$ is said to **satisfy an expectation element** $e = \langle S_e, \mathcal{O}, k \rangle$ for a given domain $\mathcal{D}$, or more concisely $e \models_{\mathcal{D}} \pi$, if the occupancy frequency for state $s$ ($x(s)$) under policy $\pi$ as evaluated for $\mathcal{D}$ satisfies the specified relation, i.e., $\mathcal{O}(\sum_{s \in S_e} x^\pi(s), k)$ is true.*

To relate these expectations to the reward specification process, we first need a way to represent the human decision-making process. For this discussion, we will start with a very abstract model of human decision-making, which we will later ground in Section 4. In particular, we will represent human decision-making using a planning function $\mathcal{P}^H$, that maps a given model to a (possibly empty) set of policies.

**Definition 3.** *For a given model $\mathcal{M}$, the set of policies that the human thinks will be selected is given by the **planning function** $\mathcal{P}^H : \mathbb{M} \to 2^\Pi$, where $\mathbb{M}$ is the space of possible models and $\Pi$ the space of possible policies.*

It is worth noticing that, as with $\mathcal{D}^H$, $\mathcal{P}^H$ is a reflection of the human's belief about the robot's model and planning capabilities. In many cases, humans might ascribe a lower level of cognitive capability to the robot. However, for this paper, we will ignore such considerations. With these definitions in place, we can describe how humans would choose an acceptable reward specification. Specifically, the human would think a reward function is sufficient if all policies they believe could be selected under this specification will satisfy their expectations, or more formally

**Definition 4.** *For a given the human's belief about the task domain $\mathcal{D}^H$ and an expectation set $\mathbb{E}^H$, a reward function $\mathcal{R}$ is considered **human-sufficient**, if for every policy in $\pi \in \mathcal{P}^H(\langle \mathcal{D}^H, \mathcal{R} \rangle)$, you have $e \models_{\mathcal{D}^H} \pi$, for all $e \in \mathbb{E}^H$.*

In this definition, we can already see the outlines of the various sources of misspecification. For one, humans use their understanding of the task and the robot's capabilities (captured by $\mathcal{D}^H$) to choose the reward specification. The true robot capabilities or task-level constraints could drastically differ from human beliefs. Next, the policies they anticipated being selected ($\mathcal{P}^H(\langle \mathcal{D}^H, \mathcal{R} \rangle)$) could be

very different from what might be chosen by the robot. There is also the additional possibility that the human is not even able to correctly evaluate whether an expectation element holds for a given policy, especially given the fact that people are known to be notoriously bad at handling probabilities [Tversky and Kahneman, 1983, Kahneman and Tversky, 1981]. In many cases, the expectations might be limited to the occupancy frequencies being forced to be zero (avoid certain states) or taking a non-zero value (should try to visit certain states). Here, the human could ignore the probabilities and reason using a determinized version of the model [Yoon et al., 2008]. Even here, there is a possibility that humans could overlook some paths that might cause the expectation element to be unsatisfied.

In general, we will deem a reward function misspecified if at least one robot optimal policy has at least one unsatisfied expectation set element.

**Definition 5.** *A human-sufficient reward function $\mathcal{R}$ is **misspecified** with respect to the robot task domain $\mathcal{D}^R$ and the human expectation set $\mathbb{E}^H$, if there exists an $e' \in \mathbb{E}^H$ and a policy $\pi$ that is optimal for $\mathcal{M}^R = \langle \mathcal{D}^R, \mathcal{R} \rangle$, such that $e' \not\models_{\mathcal{D}^R} \pi$.*

In this setting, it no longer makes sense to talk about a true reward function anymore. As far as the human is concerned, $\mathcal{R}$ is the true reward function because it exactly results in the behavior they want in *their* domain model of the task ($\mathcal{D}^H$). One can even show that there may not exist a single reward function that allows the expectation set to be satisfied in both the human and robot models.

**Theorem 1.** *There may be human and robot domains, $\mathcal{D}^H$ and $\mathcal{D}^R$, and an expectation set $\mathbb{E}^H$, such that one can never come up with a human-sufficient reward function $\mathcal{R}$ that is not misspecified with respect to $\mathcal{D}^R$, even if one allows the human planning function $\mathcal{P}^H$ to correspond to some subset of optimal policies in the human model.*

*Proof Sketch.* We can prove this by constructing an example where it holds. Consider the example discussed in the introduction. Here, there is a state that the human wants the robot to visit (i.e., the occupancy frequency $> 0$), and there is a state that the human wants to avoid(i.e., the occupancy frequency $= 0$). To achieve the expected behavior in the human model, they have to choose a reward function that will result in an optimal policy where the button they believe will lead to the goal state has to be pressed. However, if the functionality of the switches is reversed in the robot model, none of those rewards will result in a robot policy that will lead to the state that the human wanted to visit. This shows that, for this example, there exists no reward function that will result in a policy that satisfies the expectation set. Thus proving the theorem statement. $\square$

This theorem shows that one can't talk about a single 'true' reward function that holds for both domains. In the example, as far as the human is concerned, the reward they came up with is the true reward since it results in the exact behavior they wanted. In fact, any reward function that results in the robot achieving its goal will be considered incorrect since it will never lead to the expected behavior in their model. This rules out the possibility of using methods like inverse reward design [Hadfield-Menell et al., 2017], which aims to generate a single reward function that applies to both the intended and actual environments. On the other hand, the underlying expectations are directly transferrable across the domains. This brings us to what the robot's objective should be, namely, to come up with an expectation-aligned policy, i.e., one that *satisfies human expectations in the robot task domain*, when possible, or recognize when it cannot do so.

**Definition 6.** *For a human specified reward function $\mathcal{R}^H$, human expectation set $\mathbb{E}^H$, and the robot task domain $\mathcal{D}^R$, a policy $\pi^E$ is said to be **expectation-aligned**, if for all $e \in \mathbb{E}^H$, you have $e \models_{\mathcal{D}^R} \pi^E$, where $\mathcal{M}^R = \langle \mathcal{D}^R, \mathcal{R}^H \rangle$*

The primary challenge to generating expectation-aligned policies is that the expectation set is not directly available. However, unlike previous works in this space, the fact that we have a model that maps the expectations to the reward function means that there is an opportunity to develop a more diverse set of possible solution approaches. For one, we could now look at the possibility of starting by learning the human model (using methods similar to the ones described by Gong and Zhang [2020] and Reddy et al. [2018]) and use that information to produce estimates of $\mathbb{E}^H$ from the reward specification. Along the same lines, we could leverage psychologically feasible models as a stand-in for $\mathcal{P}^H$ to further improve the accuracy of our estimates (possible candidates include noisy-rational model [Jeon et al., 2020]). In the next section, we show how we can develop effective algorithms for generating expectation-aligned policies by looking at an instance where the expectation set is limited to a certain form.

# 4   Identifying Expectation-Aligned Policies

We will ground the proposed framework in a use case with a relatively simple planning function $\mathcal{P}$ that returns the set of optimal policies and where the expectation set takes a particular form. Specifically, we assume that there is an unknown set of states in the expectation set that cannot be visited, i.e., there exists some $e \in \mathbb{E}^H$, takes the form $\langle \{s\}, =, 0 \rangle$ and some states that they would like the robot to visit (i.e., $e$ of the form $\langle \{s\}, >, 0 \rangle$). We will call the states that are part of the first set of expectations the forbidden state set $\mathcal{S}^{\mathcal{F}}$ and the second one the goal set $\mathcal{S}^{\mathcal{G}}$. The expectation set described here corresponds to the widely studied reward misspecification problem type, where the robot needs to avoid negative side effects while achieving the goal (cf. Saisubramanian et al. [2022]).

Let $\mathcal{R}^H$ be the reward function specified by the user. Since there are already works on learning human beliefs about the domain, we will assume access to $\mathcal{D}^H$. The first observation we can make from the given setting is the fact that any state that can be visited by a policy that is optimal in $\mathcal{R}^H$ cannot be part of the forbidden state set (as evaluated in $\mathcal{D}^H$).

**Proposition 1.** *There exists no state $s \in \mathcal{S}^{\mathcal{F}}$ and policy $\pi \in \Pi^*_{\mathcal{M}^H}$, such that $x^\pi(s) > 0$ is true.*

The proof sketch for the proposition is provided in Appendix 8.1. Next, we can also see that the states part of $\mathcal{S}^{\mathcal{G}}$ must be reachable under every optimal policy.

**Proposition 2.** *For every state $s \in \mathcal{S}^{\mathcal{G}}$ and policy $\pi \in \Pi^*_{\mathcal{M}^H}$, $x^\pi(s) > 0$ must always be true.*

The proof sketch for the above proposition is again provided in Appendix 8.1. One might be tempted to generate the sets $\mathcal{S}^{\mathcal{F}}$ and $\mathcal{S}^{\mathcal{G}}$ by looking at all states with low and high reward values, respectively. However, there are multiple problems with such a naive approach. Firstly, it is well-known that people encode solution strategies into the reward function in addition to eventual goals [Booth et al., 2023]. This means there may be high-reward value states that are not part of $\mathcal{S}^{\mathcal{G}}$ or relatively low-reward states that are not part of $\mathcal{S}^{\mathcal{F}}$. These may have been added to get the robot to behave in certain ways. Secondly, humans may ignore states they think are guaranteed to be reached or will never be reached.

While we can't exactly calculate $\mathcal{S}^{\mathcal{F}}$ and $\mathcal{S}^{\mathcal{G}}$, we will proceed to show how we can generate supersets of these sets. In particular, we will calculate the set of all states not reachable under any optimal policies (denoted as $\widehat{\mathcal{S}}^{\mathcal{F}} \supseteq \mathcal{S}^{\mathcal{F}}$) and the set of all states that are reachable under every optimal policy ($\widehat{\mathcal{S}}^{\mathcal{G}} \supseteq \mathcal{S}^{\mathcal{G}}$). We can generate these sets by using modified forms of the LP formulation discussed in Equation 1. To test whether a given state $s_i$ is part of $\widehat{\mathcal{S}}^{\mathcal{F}}$ or not, we will look at an LP that will try to identify an optimal policy that will visit $s_i$. Specifically, the LP would look like:

$$\max_x \sum_{s,a} x(s,a) \times r(s,a) + \alpha \times (\sum_a x(s_i,a))$$
$$\text{s.t.} \quad \forall s \in S, \sum_a x(s,a) = \delta(s,s_0) + \gamma \times \sum_{s',a'} x(s',a') \times T^H(s',a',s) \tag{2}$$
$$\sum_{s,a} x(s,a) \times r(s,a) = V^*_{s_0}$$

Where $\alpha$ is some positive valued coefficient and $V^*_{s_0}$ is the value of the optimal policy in the starting state $s_0$. We can calculate $V^*_{s_0}$ by solving Equation 1 on the human model with the specified reward. By adding the new term to the objective, we provide higher objective value to policies that visit the state $s_i$. At the same time, the constraint $\sum_a x(s_0,a) = V^*_{s_0}$ ensures we only consider optimal policies. We can now formally state the following property for the above LP formulation.

**Proposition 3.** *For the LP described in Equation 2, if $s_i \in \mathcal{S}^{\mathcal{F}}$ then for the optimal value $x^*$ identified for the LP, the condition $\sum_a x^*(s_i,a) = 0$ must hold.*

This follows directly from the definitions of the expectation set and the planning function (a longer sketch is provided in Appendix 8.1). We will again employ a variation of the LP formulation to identify $\widehat{\mathcal{S}}^{\mathcal{G}}$. Specifically, to test if a given state $s_i$ is part of $\widehat{\mathcal{S}}^{\mathcal{G}}$ or not, we test the solvability of an LP that tries to create a policy optimal value that doesn't visit $s_i$.

$$\max_x \sum_{s,a} x(s,a) \times r(s,a)$$
$$\text{s.t.} \quad \forall s \in S, \sum_a x(s,a) = \delta(s,s_0) + \gamma \times \sum_{s',a'} x(s',a') \times T^H(s',a',s) \tag{3}$$
$$\sum_a x(s_0,a) \times r(s,a) = V^*_{s_0}; \quad \sum_a x(s_i,a) = 0$$

This brings us to the proposition.

**Algorithm 1** The overall query process

---

**Input:** $\mathcal{D}^H, \mathcal{D}^R, \mathcal{R}^H$
$\widehat{\mathcal{S}}^{\mathcal{F}} \leftarrow$ Calculate from $\mathcal{D}^H$ and $\mathcal{R}^H$
$\widehat{\mathcal{S}}^{\mathcal{G}} \leftarrow$ Calculate from $\mathcal{D}^H$ and $\mathcal{R}^H$
$\mathcal{S}^{\mathcal{F}}_* \leftarrow \emptyset$
$\mathcal{S}^{\mathcal{G}}_* \leftarrow \emptyset$
**while** $|\widehat{\mathcal{S}}^{\mathcal{G}} \cup \widehat{\mathcal{S}}^{\mathcal{F}}| \neq 0$ **do**
    $LP \leftarrow CreateLP(\mathcal{D}^R, \widehat{\mathcal{S}}^{\mathcal{G}}, \widehat{\mathcal{S}}^{\mathcal{F}}, \mathcal{S}^{\mathcal{G}}_*, \mathcal{S}^{\mathcal{F}}_*)$
    $Solve, X, \mathbb{D}_G, \mathbb{D}_F \leftarrow Solve(LP)$
    $\mathbb{D}^+_G, \mathbb{D}^+_F \leftarrow FindPositiveValues(\mathbb{D}_G, \mathbb{D}_F)$
    **if** $Solve$ not True **then**
        **return** No Solution
    **end if**
    **if** $|\mathbb{D}^+_G \cup \mathbb{D}^+_F| = 0$ **then**
        **return** Policy corresponding to $X$
    **end if**
    $\mathbb{S}'_{\mathcal{G}}, \mathbb{S}'_{\mathcal{F}} \leftarrow MapToStates(\widehat{\mathcal{S}}^{\mathcal{G}}, \mathbb{D}^+_G, \widehat{\mathcal{S}}^{\mathcal{F}}, \mathbb{D}^+_F)$
    $\widehat{\mathcal{S}}^{\mathcal{G}} = \widehat{\mathcal{S}}^{\mathcal{G}} \setminus \mathbb{S}'_{\mathcal{G}}, \ \widehat{\mathcal{S}}^{\mathcal{F}} = \widehat{\mathcal{S}}^{\mathcal{F}} \setminus \mathbb{S}'_{\mathcal{F}}$
    $\mathcal{S}^{\mathcal{G}}_* \leftarrow \mathcal{S}^{\mathcal{G}}_* \cup QueryGoal(\mathbb{S}'_{\mathcal{G}}), \mathcal{S}^{\mathcal{F}}_* \leftarrow \mathcal{S}^{\mathcal{F}}_* \cup QueryForbidden(\mathbb{S}'_{\mathcal{F}})$
**end while**
**return** No solution

---

**Proposition 4.** *For the LP described in Equation 3, if $s_i \in \mathcal{S}^{\mathcal{G}}$, then there must exist no solution for the given LP.*

Now with both sets $\widehat{\mathcal{S}}^{\mathcal{F}}$ and $\widehat{\mathcal{S}}^{\mathcal{G}}$ calculated, the objective of the robot is to generate a policy that covers all of the states in $\widehat{\mathcal{S}}^{\mathcal{G}}$, while avoiding the states in $\widehat{\mathcal{S}}^{\mathcal{F}}$. This may not always be possible, so we will need the system to query the user about whether a state $s \in \widehat{\mathcal{S}}^{\mathcal{F}}$ is actually part of $\mathcal{S}^{\mathcal{F}}$ (i.e., "do I need to avoid $s$?"), and if a state $s' \in \widehat{\mathcal{S}}^{\mathcal{G}}$ is actually part of $\mathcal{S}^{\mathcal{G}}$ ("do I need to visit $s'$?"). Based on their response, we can update the sets and try to see if we can now generate a policy that avoids remaining states in the updated set of forbidden states and visits all the states in the updated goal state set. We can identify whether such a policy exists and possible states to query by using the LP:

$$\max_{x, \mathbb{D}_F, \mathbb{D}_G} -1 \times \sum_{d \in \mathbb{D}_G \cup \mathbb{D}_F} d$$
$$\text{s.t.} \quad \forall s \in S, \sum_a x(s,a) = \delta(s, s_0) + \gamma \times \sum_{s', a'} x(s', a') \times T^R(s', a', s) \quad (4)$$
$$\forall d \in \mathbb{D}_G \cup \mathbb{D}_F, d \geq 0$$
$$\forall s_i \in \widehat{\mathcal{S}}^{\mathcal{F}} \sum_a x(s_i, a) - d_i = 0; \quad \forall s_j \in \widehat{\mathcal{S}}^{\mathcal{G}} \sum_a x(s_j, a) + d_j \geq 0$$

As seen in the equation, this formulation requires the introduction of two new sets of bookkeeping variables $\mathbb{D}_F$ and $\mathbb{D}_G$, such that $|\mathbb{D}_F| = |\widehat{\mathcal{S}}^{\mathcal{F}}|$ and $|\mathbb{D}_G| = |\widehat{\mathcal{S}}^{\mathcal{G}}|$. To simplify notations, we will denote the bookkeeping variable corresponding to a state $s_i \in \widehat{\mathcal{S}}^{\mathcal{F}} \cup \widehat{\mathcal{S}}^{\mathcal{G}}$ as $d_i$. The next thing to note is that all occupancy frequency is calculated using the robot model. The general idea here is that we turn the requirement of meeting reachability constraints that are part of $\widehat{\mathcal{S}}^{\mathcal{F}}$ and $\widehat{\mathcal{S}}^{\mathcal{G}}$ into soft constraints that can be ignored at a cost. This is done through the manipulation of the bookkeeping variables. This brings us to the first theorem

**Theorem 2.** *If there exists a policy that satisfies the requirement that no state in $\widehat{\mathcal{S}}^{\mathcal{F}}$ is visited and all states in $\widehat{\mathcal{S}}^{\mathcal{G}}$ is visited (therefore satisfied the underlying expectation), then the optimal solution for Equation 4, must assign 0 to all variables in $\mathbb{D}_G$ and $\mathbb{D}_F$.*

*Proof Sketch.* The reasoning follows directly from how the last two constraints of Equation 4 and the fact setting more $d$ variables to zero will result in a higher objective value. Thus, the LP would want to minimize assigning positive values to $d$ variables. It will only give $d$ a positive value if a forbidden state has a non-zero occupancy frequency or if some potential goal states are unreachable. $\square$

When such solutions are not found, the $d$ variables that take positive values tell us potential states we can query. After the query if the user says a state shouldn't be visited or should be visited the

corresponding states are removed from $\widehat{\mathcal{S}}^{\mathcal{F}}$ or $\widehat{\mathcal{S}}^{\mathcal{G}}$ and moved to the set $\mathcal{S}_*^{\mathcal{F}}$ (set of known forbidden states) and $\mathcal{S}_*^{\mathcal{G}}$ (set of known goal states). Once these sets become non-empty, we use an updated form of Equation 4, which now has hard constraints of the form.

$$\forall s_i \in \mathcal{S}_*^{\mathcal{F}}, \sum_a x(s_i, a) = 0; \ \forall s_i \in \mathcal{S}_*^{\mathcal{G}}, \sum_a x(s_i, a) \geq 0 \tag{5}$$

Theorem 2 also holds for this new LP. If the user answers that a state is neither a forbidden nor a goal state, then the state is simply removed from the corresponding superset. This process is repeated until a policy is found, if there are no more elements in $\widehat{\mathcal{S}}^{\mathcal{F}} \cup \widehat{\mathcal{S}}^{\mathcal{G}}$, or if the LP is unsolvable.

Algorithm 1 provides the pseudo-code for the query procedure. The procedure $CreateLP$ creates an LP of the form described in Equation 4, for the current estimates for $\widehat{\mathcal{S}}^{\mathcal{F}}$ and $\widehat{\mathcal{S}}^{\mathcal{G}}$ (along with known ones). The estimates are refined through queries with the user (represented by the procedures $QueryGoal$ and $QueryForbidden$). The algorithm stops if a solution is found that doesn't require any additional queries, if the LP is unsolvable, or if it runs out of states to query.

**Proposition 5.** *Algorithm 1, is guaranteed to exit in finite steps for all finite state space MDPs.*

The above proposition holds since, in the worst case, the algorithm would query about every state in the MDP. Appendix 8.2 provides an extension of the method for a planning function based on a noisy rational model.

## 5   Related Works

Given the ever-increasing capabilities of AI agents, there has been a lot of work that has looked at how to handle partially specified or misspecified objectives. Most work in this area tends to be motivated by the need to ensure safety. This is fueled by the recognition that people tend to be bad at correctly specifying their objectives [Booth et al., 2023, Milli et al., 2017], and optimizing for incorrectly specified objectives could have disastrous results [Hadfield-Menell et al., 2016]. One could roughly categorize the work done to develop more general methods to address this problem into two broad categories.

In the first, the reward function is mostly taken to be true, but they assume there is a set of hard constraints, usually partially specified, on what state features could be changed [Weld and Etzioni, 1994], which are usually referred to as side-effects. As such, these works are usually framed in terms of avoiding negative side effects. Many works in this direction focus on the problem of how to identify potential side-effect features (cf. [Zhang et al., 2018, Saisubramanian et al., 2022, Mahmud et al., 2023b]), mostly by querying the users. Recent work has also looked at extending these methods to non-markovian side-effects [Srivastava et al., 2023] and even to possible epistemic side effects [Klassen et al., 2023]. In addition to querying, there are also works that look at minimizing potential side effects when coming up with behavior [Klassen et al., 2022b]. Previous work includes work in the context of factored planning problems (cf. [Klassen et al., 2022a, Mechergui and Sreedharan, 2023]) and reinforcement learning [Vamplew et al., 2021].

The second category treats the reward function provided by the user as merely constituting a partial specification or an observation of the true reward function held by the user. A canonical example is inverse reward design [Hadfield-Menell et al., 2017], which uses the specified reward function as a proxy to infer the true objective. For a good balance between conservatism and informativeness, Krasheninnikov et al. [2021] proposes an algorithm to infer the reward function from two conflicting reward function sources. Mahmud et al. [2023a] use explanations within their framework to verify and improve reward alignment. Pan et al. [2022] proposes anomaly detection to tackle reward hacking due to misspecification. This is also related to the CIRL framework [Hadfield-Menell et al., 2016], which eschews an agent from having its own reward function, but tries to maximize the human reward function. However, the agent estimate of this reward function may also be incorrect or incomplete.

Our expectation-alignment framework allows us to capture the requirements of both of these sets of works. We can set negative side effects or safety constraints by setting the frequency of relevant states to zero. Our framework explicitly captures the fact that the specified reward should not be directly optimized in the true model. In scenarios where the reward function induces the same occupancy frequency in both human and robot models, recovering the human reward function suffices to generate expectation-aligned policies. But, as discussed this is not a general solution strategy.

Another stream of work that is relevant to the current problem is that of imitation learning [Torabi et al., 2019] or learning from demonstration [Argall et al., 2009]. Under this regime, the teacher demonstrates a particular course of action to an agent, and it is expected to either imitate or generate behaviors that match the intent behind the demonstration. Apprenticeship learning [Abbeel and Ng, 2004], and other methods that use inverse reinforcement learning (cf. [Arora and Doshi, 2021]) treat the reward function as an unknown entity and use the demonstration and any knowledge about the demonstrator to identify the true reward function. The objective then becomes to identify a policy that maximizes this learned reward function. In some sense, our work inverts this paradigm and starts from a specified reward function and tries to recreate what expectations may have led to this reward function. In addition to all the work on partially specified rewards, our work is also connected to preference/model elicitation (cf. [Boutilier, 2002] and [Chen and Pu, 2004]). Our proposed method can be thought of as a form of preference elicitation, except we focus on learning some specific kinds of preferences. It is also worth noting that there have been some efforts within the inverse-reinforcement learning community to formalize misspecification (cf. [Skalse and Abate, 2023]). However, we see such work being complementary to our effort.

## 6 Evaluation

Our primary goal with the empirical evaluation was to compare our proposed method against two baseline methods selected from the two groups of works described in Section 5. Specifically, we wanted to test:

*"How the method described in Section 4 compared with existing methods in terms of (a) computational efficiency (measured in terms of time-taken), (b) overhead placed on the user (number of queries) and (c) ability to satisfy user-expectations."*

In particular, we selected Minimax-Regret Querying (MMRQ-k) [Zhang et al., 2018] (with query size $k = 2$ and all features set to unknown) and a modified form of the Inverse Reward Design method [Hadfield-Menell et al., 2017]. To simplify the setting, we considered an MDP formulation where rewards were associated with just states as opposed to states and actions. For the modified IRD, we avoided the expensive step of calculating posterior distribution over the reward function set and instead directly used $\widehat{S}^G$ and $\widehat{S}^F$. For the expected reward function, a high positive reward was assigned for $\widehat{S}^G$ and a negative one for $\widehat{S}^F$. We then try to solve the MDP for this reward function using an LP planner. For the query-based methods, we simply check with the ground truth on whether a state belongs to the forbidden state set or to the goal state set.

**Test Domains.** We tested our method and baseline on five domains. Most of these are standard benchmark tasks taken from the SimpleRL library [Abel, 2019]. Since all examples were variations of a basic grid-world domain, we considered four different sizes and five random instantiations of each grid size (obtained by randomizing the initial state, goal state, forbidden states, and location of objects). For each of the tasks, the expectation set consists of reaching the goal state and avoiding some random states in the environment. The human models were generated by modifying the original task slightly. The walkway involves a simple grid world where the robot can use a movable walkway to reach certain states easily, but the human is unaware of it. Obstacles involve the robot navigating to a goal while avoiding obstacles (the human model includes incorrect information about the obstacles). Four rooms [Sutton et al., 1999] involves the robot navigating through connected rooms, but in the human model, the use of certain doors may not be allowed. In Puddle, the robot needs to travel to the goal while avoiding certain terrain types, while the human model may be wrong about the location of various terrain elements. Finally, in maze, the robot needs to navigate a maze, while the human model may have incorrect information about what paths are available.

**Evaluation.** All the baselines were run with a time-bound of 30 minutes per problem. All experiments were run on AlmaLinux 8.9 with 32GB RAM and 16 Intel(R) Xeon(R) 2.60GHz CPUs. We used CPLEX [Bliek1ú et al., 2014] as our LP solver (no-cost edition)[4]. First, *we found that the MMRQ-k method could not solve any of the instances*. This was because the runtime was dominated by the time needed to calculate the exponential number of dominant policies. This shows how computationally

---

[4]The code for our experiments can be found at `https://github.com/Malek-Mechergui/codeMDP`

| Problem Instance | | Our method | | IRD | |
| --- | --- | --- | --- | --- | --- |
| Domain | Grid size | Query Count | Time (secs) | No of Violated Expectations | Time (msecs) |
| Walkway | (4,4) | $2.2 \pm 0.83$ | $7.7 \pm 0.4$ | $1.6 \pm 0.5$ | $32.3 \pm 1.4$ |
| | (5,5) | $3.2 \pm 1.48$ | $11.8 \pm 0.45$ | $2.26 \pm 1.03$ | $45 \pm 1.12$ |
| | (9,9) | $4.5 \pm 4.8$ | $60 \pm 1$ | $2.5 \pm 3$ | $215.7 \pm 3$ |
| | (11,11) | $14.25 \pm 2.91$ | $138.57 \pm 6.32$ | $3.6 \pm 3.9$ | $456.05 \pm 19$ |
| Obstacles | (4,4) | $2.4 \pm 1.34$ | $9.6$ | $1.4 \pm 0.58$ | $32.4 \pm 1.27$ |
| | (5,5) | $3 \pm 1$ | $13 \pm 0.3$ | $2 \pm 1.12$ | $46.13 \pm 2.42$ |
| | (9,9) | $4.5 \pm 3.1$ | $62.8 \pm 0.9$ | $4 \pm 2.66$ | $216.66 \pm 12.4$ |
| | (11,11) | $11.75 \pm 2.87$ | $134.51 \pm 6.88$ | $6 \pm 3.7$ | $450 \pm 26.6$ |
| Four Rooms | (5,5) | $1.4 \pm 0.55$ | $3.1 \pm 1$ | $1.42 \pm 0.75$ | $49 \pm 1.7$ |
| | (7,7) | $1.8 \pm 0.45$ | $12.9 \pm 0.7$ | $1.07 \pm 0.73$ | $111.5 \pm 3$ |
| | (9,9) | $2.75 \pm 0.5$ | $71.2 \pm 0.8$ | $1.35 \pm 0.74$ | $251 \pm 3.5$ |
| | (12,12) | $4.44 \pm 2.065$ | $224.98 \pm 6.84$ | $1 \pm 0.57$ | $728 \pm 5.22$ |
| Puddle | (5,5) | $3 \pm 2$ | $13.2$ | $1.13 \pm 0.63$ | $49.5 \pm 1.48$ |
| | (7,7) | $5 \pm 3.46$ | $31.7 \pm 7$ | $1.2 \pm 0.63$ | $123.27 \pm 15$ |
| | (9,9) | $3.14 \pm 2.6$ | $42.11 \pm 1.86$ | $0.9 \pm 0.66$ | $275.2 \pm 5.9$ |
| | (11,11) | $2.44 \pm 1.85$ | $132.5 \pm 7.132$ | $1.3 \pm 0.6$ | $566.1 \pm 7.8$ |
| Maze | (3,3) | $1.66 \pm 0.57$ | $5.9 \pm 0.1$ | $1 \pm 0.87$ | $25.1 \pm 1.51$ |
| | (5,5) | $1.66 \pm 0.57$ | $13 \pm 1$ | $1.36 \pm 0.67$ | $47.3 \pm 1.3$ |
| | (7,7) | $2.33 \pm 0.57$ | $30 \pm 0.5$ | $1.46 \pm 0.66$ | $106 \pm 1.6$ |
| | (9,9) | $7.5 \pm 6.27$ | $35.34 \pm 14.18$ | $1.42 \pm 0.51$ | $244.29 \pm 14$ |

Table 1: For our method, the table reports the number of queries raised and the time taken by our method. For IRD, it shows the number of expectations violated by the generated policy and the time taken. Note that our method is guaranteed not to choose a policy that results in violated expectations.

expensive methods from the first group are. On the other hand, our method is much faster, and even in the largest grids, it took less than five minutes (Table 1) and only required very few queries. Note that the maximum number of queries that could be raised in each case corresponds to the total state space. In each case considered here the number of queries raised was substantially smaller than the state space size. Thus showing the effectiveness of our method compared to existing query methods.

In terms of comparing our method to IRD, the main point of comparison would be a number of violated expectations (after all IRD doesn't support querying). Table 1 shows how, in almost all the cases, IRD generated policies that resulted in expectation violations. On the other hand, our method guarantees policies that will never result in violation of user policies. While IRD is fast, please keep in mind that we avoided the expensive inference process of the original IRD with direct access to $\mathcal{S}^{\mathcal{F}}$ and $\mathcal{S}^{\mathcal{G}}$ (reported times don't include the calculation of these sets).

# 7 Conclusion

This paper introduces a novel paradigm for studying and developing approaches to handle reward misspecification problems. The *expectation-alignment* framework takes the explicit stand that any reward function a user provides is their attempt to drive the agent to generate behaviors that meet some underlying expectations. We formalize this intuition to explicitly define reward misspecification, and we then use this framework to develop an algorithm to generate behavior in the presence of reward misspecification. We empirically demonstrate how our method provides a significant advantage over similar methods to handle this reward misspecification in standard MDP planning benchmarks.

**Limitations.** One aspect of reward misspecification we haven't discussed here is the one caused by human errors during transcription or communication of the reward function (say, bugs in the reward function code). As previous works have shown [Booth et al., 2023], reward misspecification is quite frequent in practice, even in the absence of such errors. This paper also focuses on settings where the models and planning functions are known upfront. We believe our method sets up a solid formal framework that can be used as a basis to develop future RL methods that could relax these assumptions. The paper also only instantiates a specific form of expectation set. More work needs to be done to identify when different forms of expectation may be appropriate. Not only could different forms of expectation set be more intuitive or natural for different settings, but it might also impact how effectively the user can be queried.

## Acknowledgments and Disclosure of Funding

Sarath Sreedharan's research is supported in part by grant NSF 2303019. This material is based in part upon work supported by Other Transaction award HR00112490377 from the U.S. Defense Advanced Research Projects Agency (DARPA) Friction for Accountability in Conversational Transactions (FACT) program. Approved for public release, distribution unlimited.

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

# 8 Appendix / supplemental material

## 8.1 Proof Sketch for Propositions

**Proposition 1.** *There exists no state $s \in \mathcal{S}^{\mathcal{F}}$ and policy $\pi \in \Pi^*_{\mathcal{M}^H}$, such that $x^\pi(s) > 0$ is true.*

*Proof Sketch.* We will prove this through contradiction. Let's assume that there exists a state $s \in \mathcal{S}^{\mathcal{F}}$, where $x^\pi(s) > 0$ for an optimal policy $\pi \in \Pi^*_{\mathcal{M}^H}$ for a human specified reward function $\mathcal{R}^H$. Per Definition 4, a reward function is only specified, i.e., they are human-sufficient if, for every policy in $\pi \in \mathcal{P}^H(\langle \mathcal{D}^H, \mathcal{R} \rangle)$, you have $e \models_{\mathcal{D}^H} \pi$, for all $e \in \mathbb{E}^H$. If $s \in \mathcal{S}^{\mathcal{F}}$, then $e = \langle \{s\}, =, 0 \rangle$. Per our assumptions, $\mathcal{P}$ returns the set of optimal policies, hence $\pi \in \mathcal{P}^H(\langle \mathcal{D}^H, \mathcal{R}^H \rangle)$. This means that $e \models_{\mathcal{D}^H} \pi$, which is only true if $x^\pi(s) = 0$. This contradicts our initial assertion, hence proving our statement by contradiction. $\square$

**Proposition 2.** *For every state $s \in \mathcal{S}^{\mathcal{G}}$ and policy $\pi \in \Pi^*_{\mathcal{M}^H}$, $x^\pi(s) > 0$ must always be true.*

*Proof Sketch.* We can prove this through contradiction by following a rationale similar to the earlier proposition. We will again leverage the intuition that for a reward function to be human-sufficient, all optimal policies must satisfy all specified expectations, which include visiting states in $\mathcal{S}^{\mathcal{G}}$. As such, if a state is not visited by an optimal policy, it cannot be part of the set $\mathcal{S}^{\mathcal{G}}$. $\square$

**Proposition 3.** *For the LP described in Equation 2, if $s_i \in \mathcal{S}^{\mathcal{F}}$ then for the optimal value $x^*$ identified for the LP, the condition $\sum_a x^*(s_i, a) = 0$ must hold.*

*Proof Sketch.* In the equation the constraint $\sum_{s,a} x(s,a) \times r(s,a) = V^*_{s_0}$, ensures that the identified occupancy frequency corresponds to an optimal policy. The LP is trying to maximize the objective function. Since the first term of the objective function $\max_x \sum_{s,a} x(s,a) \times r(s,a)$ corresponds to the value of the policy, it must be equal for all optimal policies. As such, the solver will try to identify occupancy frequencies that try to maximize the occupancy frequency for visiting $s_i$. In other words, it will try to find optimal policies that visit the state. However, per Proposition 1, no states in $\mathcal{S}^{\mathcal{F}}$ are ever visited by an optimal policy. This means if the LP finds a solution with non-zero occupancy frequency for the state $s_i$ it cannot be part of $\mathcal{S}^{\mathcal{F}}$. $\square$

**Proposition 4.** *For the LP described in Equation 3, if $s_i \in \mathcal{S}^{\mathcal{G}}$, then there must exist no solution for the given LP.*

*Proof Sketch.* The proof closely resembles the one provided for the earlier proposition. The constraint $\sum_{s,a} x(s,a) \times r(s,a) = V^*_{s_0}$ forces the LP solver to only consider occupancy frequencies for optimal policies. The new constraint $\sum_a x(s_i, a) = 0$, forces the LP to find optimal policies where the state is not visited. If in fact such a policy is found, per Proposition 4, the state cannot be part of $\mathcal{S}^{\mathcal{G}}$. $\square$

## 8.2 Noisy Rational Model

It is well known that humans are better approximated as bounded rational agent, as opposed to rational agents. As such, except in the simplest scenarios, human users may not be able to identify the optimal decision. A popular model for approximating human decision-making is to use the noisy rational model Jeon et al. [2020]. Under this approach, the likelihood of the human choosing a policy $\pi$ is given as

$$p(\pi) \propto e^{\beta \times V^\pi(s_0)}$$

Where $V^\pi(s_0)$ corresponds to the value associated with state $s_0$ under a policy $\pi$. For any non-zero $\beta$ value, the user is more likely to select policies with higher values. However, even non-optimal policies have a probability of getting picked. Additionally, as the value of $\beta$ reduces, the likelihood of a non-optimal policy being selected increases, with $\beta = 0$ corresponding to a case where the user selects the policy completely randomly. As such, $\beta$ is sometimes referred to as the rationality parameter.

For a non-zero $\beta$ value, let $V^{p_i}_{S_0}$ be the value for which the noisy rational model assigns a probability of at least $p_i$ of being picked. Now, let us consider a case where we want to extend our formulation to support a planning function where the human could pick some policy with a probability greater than $p_i$. In this case, we only need to update the LP to calculate the supersets. However, here $\widehat{\mathcal{S}}^{\mathcal{F}}$ consists

of all states avoided by at least one policy and $\widehat{\mathcal{S}}^{\mathcal{G}}$ consists of every state visited by at least one policy. The corresponding LP formulation for $\widehat{\mathcal{S}}^{\mathcal{F}}$ become

$$\max_x \sum_{s,a} x(s,a) \times r(s,a)$$

$$\text{s.t.} \quad \forall s \in S, \sum_a x(s,a) = \delta(s,s_0) + \gamma \times \sum_{s',a'} x(s',a') \times T^H(s',a',s) \tag{6}$$

$$\sum_a x(s_0,a) \times r(s,a) > V_{S_0}^{p_i}, \quad \sum_a x(s_i,a) = 0$$

A state belongs to $\widehat{\mathcal{S}}^{\mathcal{F}}$ if the above LP is solvable for a given state.

$$\max_x \sum_{s,a} x(s,a) \times r(s,a) + \alpha \times \left( \sum_a x(s_i,a) \right)$$

$$\text{s.t.} \quad \forall s \in S, \sum_a x(s,a) = \delta(s,s_0) + \gamma \times \sum_{s',a'} x(s',a') \times T^H(s',a',s) \tag{7}$$

$$\sum_a x(s_0,a) \times r(s,a) > V_{S_0}^{p_i}$$

A state belongs to $\widehat{\mathcal{S}}^{\mathcal{G}}$ if the above LP is solvable for a given state.

As you can see, basically for this new planning function, we only need to swap the LP formulations, change the cost requirement to a looser one, and then check for solvability instead of unsolvability. Since we still get supersets, we can still use the pseudocode provided earlier.

## 9 Broader Impact

We see the problems and challenges being discussed here as being core to the aim of developing effective AI systems that can interact and work with people. We believe that as AI systems become more powerful, it is important that we have measures to detect possible misspecifications of rewards and objectives. The problems discussed here are also related to the problem of value alignment. However, our goal is only to develop methods to ensure AI system behavior aligns with user expectations/intentions. We are not in any way assigning whether or not user behaviors align with larger societal values. We believe this is an orthogonal problem, which would still benefit from the methods we develop.

