# OpenReview forum: "Expectation Alignment: Handling Reward Misspecification in the Presence of Expectation Mismatch"
_NeurIPS.cc/2024/Conference — NeurIPS 2024 poster_

### Official Review · Reviewer_ahA3 · 2024-06-16

**Soundness:** 4
**Presentation:** 3
**Contribution:** 4
**Rating:** 7
**Confidence:** 4

**Summary:**

This paper studies the process of reward design and as a result reward misspecification when a human designer has potentially misspecified beliefs about the robot's operating domain or generally trouble with designing a reward leading to the robot generating desired behavior in its own domain. The paper formalizes the notion of "expectation alignment" over occupancy frequency under human and robot domains satisfying a set of desired or forbidden states. Since the human's expectation set is unknown, the authors propose a set of linear programs 1) to test whether a state belongs to the forbidden set, 2) to test whether a state belongs to the goal set, and 3) to find a minimal set of queryable states by minimizing the number of states in the forbidden and goal sets which can be used to query a human designer. The proposed method is mainly compared against inverse reward design.

**Strengths:**

**Originality & significance**: This work is original to my knowledge and the proposed method of tackling reward design and specification from the perspective of occupancy measure is novel and interesting. Given the increasing popularity and maturity of dual RL, the propose method has the potential to be extended to more practical settings.

**Quality & clarity**: The paper is well written. Even though the problem studied is novel and somewhat "niche", the authors did a good job walking the readers through a lengthy problem formulation. I appreciate the thoughts that went into this problem formulation.

**Weaknesses:**

**Clarity**: I think the authors could provide more context (in main text or appendix) on inverse reward design (IRD) for readers to better understand and interpret the evaluation results. Currently, readers have to read the IRD paper to achieve that.

The authors claim that "IRD generated policies that resulted in expectation violations. On the other hand, our method guarantees policies that will never result in violation of user policies". As far as I understand, IRD proposes a model of how human specified reward relates to the true intended reward. In contrast to IRD which is a one-off process (one-shot learning if you will) where the human designer is queried only once and then the robot is deployed in the test environment, the proposed method is iterative where the human can potentially provide multiple feedback to the designed reward. In some sense, this is equivalent to assuming privileged access the test environment and the number of states in $\mathbb{D}_{F}$ after the first human query is in some sense similar to the number of violated expectations. So stating that the proposed method never violates user policies doesn't seem appropriate, or at least comparing with IRD on this metric doesn't seem to be "comparing apples to apples".


Another minor suggestion is to put equation numbers in corresponding lines in the algorithm to make it easier for readers to understand.

**Questions:**

I have some minor questions on LP formulation and notation:
* In eq 2 and 3, is the notation $V_{s_0}^*$ the optimal value function in $(D^{H}, R^{H})$ (since the authors only said calculate $V_{s_0}^*$ but did not say what is being calculated)? If so then I doubt that notation $\sum_{a}x(s_{0}, a) = V_{s_0}^*$ is correct in general, since $x(s, a) \in [0, 1]$ is the visitation frequency $\frac{1}{H}\mathbb{E}[\sum_{H}Pr(s_t=s, a_t=a)]$ rather than a value function.
* In all LP equations, there seem to be a typo on the transition matrix, i.e, it should probably be: $\sum_{a}x(s, a) = \delta(s, s_0) + \gamma\sum_{s', a'}x(s', a')T(s', a', s)$. See [Sikchi et al, 2024](https://arxiv.org/abs/2302.08560) eq 4.
* For the LPs in eq 2 and 3, do you have to solve it for every $s_i$ being tested? This seems expensive?
* In line 249, there seems to be a typo. I guess the authors are trying to say: "calculate the set of all **forbidden** states reachable" and "calculate the set of all **goal** states reachable".

**Limitations:**

Acknowledgement of limitations seem appropriate.

---

> ### Author Rebuttal · Authors · 2024-08-06
>
> We thank the reviewer for all the constructive feedback and for catching the typos. We will incorporate them into the final draft. Below, we have provided responses to specific questions and comments raised.
>
> IRD: We will make sure to provide a more clear description of IRD in the evaluation section. The reviewer is right in the fact that the method isn’t explicitly designed to avoid constraint violation. However, we believe that this also comes from the stance that the paper takes that there exists a single true reward function that is transferable across domains. We believe that if it had been accounted for, the authors would have included a querying strategy. We realize that this might not always be a fair comparison, and this is the reason why we also compared it with a method that explicitly performs queries. As we can see, that method cannot scale up to the problems we considered in this paper.
>
> $V^*_{s_0} $ and constraint: Thank you so much for catching it. This was an unfortunate typo that got copied around in the LP descriptions in the paper. The constraint should have read $\sum_{s,a} x(s,a)r(s) = V^*_{s_0}$. Note that $\sum_{s,a} x(s,a) r(s)$ returns the value of the state $s_0$ for the current policy (cf. [Poupart, 2005]). We will make sure to fix it in the final draft.
>
> Transition Function: We believe our formulation is correct in this case. Looking at the other paper, we believe the difference comes from how they represent the transition function. They seem to be using a conditional probability notational scheme, where they represent the probability of transition to a state s’ when action a is executed in state s as P(s’|s, a). We, on the other hand, use a simpler functional notation of the form $T: S \times A \times S \rightarrow [0,1]$. In our case, the same probability will be returned by the arguments T(s, a, s’). We hope this clears the confusion, and we will make sure to emphasize this in the background section where we define our notations.
>
> Queries: The reviewer is correct in that we have to check it against every state. However, as discussed in the main rebuttal response, as we move to larger problem settings, we expect to use factored/feature-based representations. These allow for a relatively small set of features to represent large state spaces. Under these scenarios, the tests only need to be run once per each feature. This would dramatically cut down on the number of times it needs to be run. Standard planning benchmarks use relatively small feature counts[ext1], which number less than a hundred.
>
> Lines 248-249: This was again a typo. $\widehat{\mathcal{S}}^\mathcal{F}$ corresponds to states that are not reachable under any optimal policy. We will make sure to fix it in the final draft.
>
> [ext1] International Planning Competition, IPC Competition Domains. https://goo.gl/i35bxc, 2011.

---

> > ### Comment · Reviewer_ahA3 · 2024-08-08
> >
> > I thank the authors for their responses. Most of my questions are resolved.
> >
> > I have one follow up question on the transition function. My original question is on the summands of the transition function, i.e., the current states and actions should be summed out as opposed to the future states and actions. I think this makes sense because the basic idea of the Bellman flow equation, similar to the Bellman equation, is that the long term occupancy is the sum of the immediate occupancy and the expected next occupancy. Another point of reference is eq 3 in [this paper](https://arxiv.org/abs/1906.04733) which illustrates the same point. Please feel free to correct me if I'm wrong about this.

---

> > > ### Author Response · Authors · 2024-08-08
> > > **Re: Transition Function**
> > >
> > > You are actually correct. We apologize for missing that. During the rebuttal, we were mainly focused on the form of the transition function, and we didn't notice the summand was switched. In fact, to avoid further confusion, we will stick with the convention of s' being the next state and rewrite the constraint to
> > >
> > > $\sum_{a'} x(s',a') = \delta(s',s_0) + \gamma\times \sum_{s,a} x(s,a) \times T^R(s, a, s')$
> > >
> > > This is the more popular way of denoting occupancy frequency equation (also used by the paper, the reviewer pointed to). We thank the reviewer for catching that, and we apologize for the confusion.

---

> > > > ### Comment · Reviewer_ahA3 · 2024-08-08
> > > >
> > > > I thank the authors for confirming. All my questions are resolved.

---

> > > > > ### Author Response · Authors · 2024-08-08
> > > > > **Re: Comments**
> > > > >
> > > > > Thank you again for your extremely helpful comments. We really appreciate it.

---

### Official Review · Reviewer_pegd · 2024-07-05

**Soundness:** 2
**Presentation:** 4
**Contribution:** 2
**Rating:** 7
**Confidence:** 3

**Summary:**

The paper addresses the problem of reward misspecification. It introduces a framework (EAL) to capture how humans go from setting expectations about a problem to specifying the reward for it. The problem is modeled as a single Human-Robot interaction.
After introducing the formalism, the authors propose an algorithm to solve the EAL problem. It works by mapping the inference problem about the user expectations to LPs and by obtaining queries to the human, providing an efficient and effective way of inferring user expectations given a specified reward.

**Strengths:**

- The paper is extremely well written. It introduces previous concepts clearly. The motivations for choices in the formalism introduced are also well-justified and adequately compared with the previous literature on the topic
- The theoretical contributions are very solid and sound.
- The formulation via occupancy measures and LP simplifies the problem significantly in my opinion, making it easy to understand why and how the algorithm is derived.

**Weaknesses:**

- The main weakness I find is the limited experimental evaluation. I understand that the contributions of this work are mainly theoretical, but I still think it would benefit from additional experiments

**Questions:**

- Do you have any formal proof ( to put in the Appendix) both for the Theorems and the Propositions? Even if most of the results and proof are pretty straightforward, I would still like to see the proofs formally written out
- Could you please clarify how one should read the results from Table 1? I do not directly see how to relate query count and No of Violated Expectations. If the comparison is 1 to 1, it seems your algorithm is actually performing sub-optimally?
- How do you think your algorithm can scale in non-grid environments? What about grid environments with a computationally untractable number of states? I imagine obtaining good results would require a too-high number of queries.

**Limitations:**

No ethical limitations

---

> ### Author Rebuttal · Authors · 2024-08-06
>
> We thank the reviewer for all the comments. We will make sure to incorporate them into the paper. Below, we have provided responses to some specific concerns and questions raised in the review.
>
> Formal Proof: We will be more than happy to include detailed formal proof for all the propositions and theorems. Since the allowed pdf isn’t supposed to include text, we are including a proof sketch here for proposition 1 as a sample proof.
>
> Proposition 1 -   There exists no state $s \in \mathcal{S}^\mathcal{F}$ and policy $\pi  \in \Pi^*_{\mathcal{M}^H}$, such that $x^\pi(s) > 0$ is true.
>
> Proof Sketch -
> We will prove this through contradiction. Let's assume that there exists a state $s \in \mathcal{S}^\mathcal{F}$, where $x^\pi(s) > 0$ for an optimal policy $\pi \in \Pi^*_{\mathcal{M}^H}$ for a human specified reward function $\mathcal{R}^H$. Per Definition 4, a reward function is only specified, i.e., they are human-sufficient, if for every policy $\pi \in \mathcal{P}^H(\langle \mathcal{D}^H, \mathcal{R}\rangle)$, you have $e \models_{\mathcal{D}^H} \pi$, for all $e \in \mathbb{E}^H$. If $s \in \mathcal{S}^\mathcal{F}$, then $e = \langle \{s\},=,0\rangle$. Per our assumptions, $\mathcal{P}$ returns the set of optimal policies, hence $\pi \in \mathcal{P}^H(\langle \mathcal{D}^H, \mathcal{R}^H\rangle)$. This means that $e \models_{\mathcal{D}^H} \pi$, which is only true if $x^\pi(s) = 0$. This contradicts our initial assertion, hence proving our statement by contradiction.
>
> We will include detailed proofs for all the theoretical assertions in the final draft.
>
> Experiments:
>
> As mentioned in our response to other reviewers and the main rebuttal, our paper included two baselines. One of these was another query method, which unfortunately couldn’t solve any of the problems within the given time limit, showing the scalability of our method.
>
> Table1: Please note that the violations listed by the policy identified by IRD are not the whole set of constraints, even though the reward tried to penalize potential unsafe states and reward potential goal states. These constraints are upper bound by the actual number of constraints that were present in the instances, which is always much smaller than the total number of states. On the other hand, the query method could query about a number of states that are not actual constraint states. For the queries, the worst-case upper bound is the set of all states. This again points to some of the challenges faced by previous query methods. It is also worth noting that in safety-critical settings, even violating a single constraint could be bad. So, instead of the number of violations, the bigger problem is that it violates any constraints at all.
>
> Non gridworld domains: As discussed in the main rebuttal text, we can handle domains with large state space using feature-based representations, which are already popular in the safety literature and planning in general. Here, there would be a set of features that characterize the goal states and another set that captures the states to be avoided. A feature set can capture an exponential number of states. Here, the algorithm changes will be minimal, and instead of querying over the states, we will query over features, which will keep the query count small.

---

> > ### Comment · Reviewer_pegd · 2024-08-08
> > **Thanks for your answer**
> >
> > Thanks for your response.
> >
> > **Experiments**: Thanks for clarifying your results. I suggest using this answer to increase the clarity of the presentation in the Experiments section
> >
> > **Non gridworld domains**: I've read the main rebuttal text as well, and this clarifies my main doubts.
> >
> > **Proofs**: Thanks for the sketch of the proof.
> >
> > The authors addressed most of my concerns. Including a formal proof for all the statements (even in the appendix) will much increase the quality and strength of the submission. Since the authors mentioned that this will be done, and a sample proof is provided, I'll increase my score accordingly.

---

> > > ### Author Response · Authors · 2024-08-08
> > > **Re: Reviewer Response**
> > >
> > > Thank you for the quick response and all the helpful feedback. We will make sure to incorporate them into the paper.

---

### Official Review · Reviewer_RJbh · 2024-07-13

**Soundness:** 2
**Presentation:** 3
**Contribution:** 3
**Rating:** 5
**Confidence:** 3

**Summary:**

The paper tackles the problem of reward misspecification in settings where humans have potentially incorrect beliefs about the environment. Instead of treating a true human reward function as the fundamental object, they introduce *expectation sets*, which specify the states that the human does or doesn't expect an optimal policy to visit. In a specialized setting where preferences consist purely of goal states and forbidden states, they then show how to find policies that meet those human expectations.

**Strengths:**

- Reward misspecification is an important topic, and this paper provides an interesting new perspective on it
- The ideas are clearly described and mostly easy to follow
- The expectation alignment framework/technique could be useful at least in certain settings

**Weaknesses:**

- I don't think the paper sufficiently demonstrates that the expectation alignment perspective is broadly better than thinking about an (unknown) true human reward function. Starting at line 203, the paper argues that expectation sets transfer better to different transition functions than reward functions do. This doesn't seem true in full generality: for both reward functions and expectation sets, we could assume that they express the true human preferences about states (in which case they'd transfer), or that they are entangled with the transition function. The latter could also be the case for expectation sets; for example, the human might expect the optimal policy to spend time in some actually suboptimal state simply because they incorrectly believe this is necessary to reach the goal state.
- The setting for which the paper describes an algorithm is quite limited (with only "goal states" that should be reached with non-zero probability, and forbidden states that should be reached with zero probability).
- It's hard to draw clear conclusions from the experiments:
	- The experiments measure the number of human expectations that inverse reward design violates, and compare that to the guarantee of expectation alignment not to violate any expectations. But of course, expectation alignment was directly designed for that purpose, whereas IRD was not—from the IRD perspective, the relevant thing to compare would be the reward achieved under the expected reward function that IRD's inputs are based on.
	- It would also be nice to have experiments in larger and more varied environments than just 11x11 gridworlds.

**Questions:**

Is there a clear reason to expect expectation sets to transfer better than reward functions between domains? In particular, why don't they face similar problems along the lines I mention above? i.e.:
> the human might expect the optimal policy to spend time in some actually suboptimal state simply because they incorrectly believe this is necessary to reach the goal state


As a minor note, in definition 3, should the planning function map to the *powerset* of the space of policies? That seems to be how it's used in definition 4, where the planning function evaluated on a single model is a *set* of policies rather than a single policy.

**Limitations:**

The method assumes knowledge of a way to map between states in the human and robot model, as well as knowledge about the full human and robot models. These assumptions are made in other existing work as well, so are not a damning issue. But they still seem worth highlighting given how limiting they are for many applications.

---

> ### Author Rebuttal · Authors · 2024-08-06
>
> We thank the reviewer for all the feedback. We will make sure to incorporate them into the paper. Below, we have provided responses to some specific concerns and questions raised in the review.
>
> Transference: We want to thank the reviewer for all their constructive comments. In regards to the question of transferability, our main argument is built on the fact that the expectation set constitutes the behavior they want to see the robot perform. As such, instead of the question of transference, it becomes one of whether or not the AI agent/robot can achieve the desired/expected behavior. Which our method can determine. Our claim is further supported by the abundance of evidence from cognitive science and psychology (cf. [Simon,1977]), which shows that people inherently reason about tasks and plans in terms of goals of achievement. As discussed in the paper, the expectation set, as formulated in the paper, is a generalization of goals. On the other hand, we are unaware of any evidence that shows people have intrinsic reward functions associated with tasks. Some works show that even experts struggle to design reward functions (cf. [Booth et al., 2023]). As such, the use of reward functions becomes a means to the end of specifying the behavior they want to see. However, the reviewer is correct in asserting that users might not be fully aware of what is achievable in the environment (because of knowledge or inferential limitations). In such cases, we would argue that the system should use other mechanisms like explanation to inform the user what is possible rather than determining what they should want. We will make sure to update the text to make sure that this point is clearly articulated.
>
> Specific Instance: As mentioned, our choice of specific instance was motivated by the importance of goals in human reasoning, per existing literature, and all the AI-safety works that have focused on avoiding side effects. Our current method captures both these considerations.
>
> Experiments: We chose to highlight IRD as one of the baselines because they center rewards and overlook the fact that the reward is an expression of some true underlying behavioral expectations. For the user, the reward itself is a means of achieving some behavior, and in the case we consider, where there are states to be avoided and achieved, violating them would render the policy useless. It is also worth noting that while, in the end, IRD settles for the use of an expected value, they do use behavior generated by a reward function as the means to identify potential hypotheses for true reward function. It is also worth noting that IRD was not the only baseline. We also used a query-based baseline. Unfortunately, it could not solve any of the problems we considered in the given time limit. As such, it was left out of the table, but we mentioned it in the text. This highlights the efficiency of our method and the advantages provided by considering the human model and planning set. Regarding scalability in non-grid environments, as mentioned in the main response, the primary method we can leverage is to use a feature-based representation. This representation scheme allows us to capture large state sets using a small set of features. This was briefly alluded to in the discussion about rewards. We also talk about how a large number of current work in avoiding negative side-effects already makes use of features. The occupancy frequency can easily be extended to capture the occupancy frequency of features, and the expectation set can be captured in terms of those. Here, the expectation set will consist of features to be achieved and avoided. In turn, we can update the LP formulations to use features, and constraints are represented using feature occupancy frequency. The queries will also be in terms of the features, which should keep the total number of queries fairly small. This should allow our methods to be scaled more easily. The availability of powerful and efficient LP solvers also makes our methods inherently more scalable.
>
> Powerset Defn3: Yes, this was a typo. We will fix it in the final draft.
>
> Limitation: We agree about the access to the mapping, but works exist that provide methods for learning such mappings.

---

> > ### Comment · Reviewer_RJbh · 2024-08-08
> >
> > Thank you for the detailed response! I still think it would be good to have more concrete arguments that expectation sets transfer better (or have other advantages). Everything you mention (connections to cognitive science, the difficulty of specifying reward functions) is great *motivation* for studying alternative frameworks. But I think it falls short of being sufficiently convincing evidence, given that these feel like very high-level conceptual arguments. For example:
> >
> > > the expectation set, as formulated in the paper, is a generalization of goals
> >
> > Yes, but we could also think of reward functions as a generalization of goals (where goals would be encoded as reward functions that only give rewards 0 or 1). Clearly, expectation sets generalize goals in an interestingly different way, and maybe a better way. But whether (or in which cases) expectation sets are the "right" generalization of goals seems like the crucial question.
> >
> > > However, the reviewer is correct in asserting that users might not be fully aware of what is achievable in the environment (because of knowledge or inferential limitations). In such cases, we would argue that the system should use other mechanisms like explanation to inform the user what is possible rather than determining what they should want.
> >
> > I don't think I follow this argument. If we assume that the system can explain everything the user needs to know about the environment, this just seems to solve the problem I understood the paper to be tackling. (Humans having incorrect beliefs about the environment and thus providing misspecified rewards.) In this setting, where humans now have correct beliefs about the environment, it seems reward functions should work just as well as expectation sets. (At least in the framework presented in the paper: if the human model and robot models match, then every human-sufficient reward function should be correctly specified, I think?)
> >
> > So it seems to me that either we don't assume the robot can explain everything to the human, in which case both reward functions and expectation sets can fail to transfer in at least some cases, or we let the robot explain everything, in which case both approaches work fine. It might still be the case that expectation sets transfer more often or have some other advantage, but I currently don't see the concrete argument in favor of that claim.
> >
> > Please let me know in case I've misunderstood anything!
> >
> > To be clear, I think it would be unreasonable to expect this paper to fully demonstrate that expectation sets are always the right approach. But I do think it would be good if there was at least one clear demonstrable advantage of expectation sets, rather than only high-level arguments (that in my view are only moderately compelling, but naturally views on that will diverge more than they would for more specific claims).

---

> > > ### Author Response · Authors · 2024-08-08
> > > **Re: Response to the reviewer**
> > >
> > > Thank you for the quick response.
> > >
> > > The main difference is that the expectation set is a direct encoding of agent behavior. The occupancy frequency can be though of as being determined by the traces the policy would generate over the current transition function. As such, it is already accounting for the transition function and what will happen. However, one needs to use the transition function to figure out what policy would be generated in response to that reward function. Thus, the question of transferability never arises in our case because the users want to see the robot perform the behavior they expected (another way to think about it is that the expectation set is directly transferred over without any modifications). They want to see the robot follow a policy that meets the expectations defined over the occupancy frequency in the robot model. The question that arises is whether the robot can achieve this behavior or not, which our method directly tries to address.
> > >
> > >
> > > It is also worth repeating that the two main central assertions in this paper are as follows:
> > >
> > > 1. People don't necessarily start with a reward function. Rather, it is more plausible that they have some target behavior in mind that they want the robot/AI agent to perform.
> > >
> > > 2. The human and robot models could be different. This is, in fact, a very common case. Limited situational awareness, i.e., cases where the user's understanding of the task may be wrong, is a popular issue studied within human factors/psychology literature. The unfortunate fact is that as the complexity of the task increases, it might not be possible to completely resolve this discrepancy. This is a fact that is accepted by most explainable AI methods. This is one of the reasons why abstractions are a very popular tool within XAI.
> > >
> > > Please let us know if this answers your questions. We would be more than happy to expand on any of the points discussed above.

---

> > > > ### Author Response · Authors · 2024-08-09
> > > > **Follow up on the comment**
> > > >
> > > > We hope our response helped answer your question. If not, please let us know; we would be happy to elaborate.

---

> ### Comment · Reviewer_RJbh · 2024-08-09
>
> Thanks for the additional explanation; this has helped clarify some things for me. I have two key uncertainties left, first about the intended scope and second about Theorem 1 (or more broadly about examples of expectation sets being a better perspective than rewards, but Theorem 1 seems a good candidate for that).
>
> **Scope:** To make this concrete, let me describe a toy example I've been thinking about. Say we have a gridworld with a fixed start and goal state (for simplicity), and the human wants the robot to move to the goal as quickly as possible. The human (incorrectly) believes that the geometrically shortest path from start to goal is free of undesirable states. So the policy they expect is to walk directly from the start to the goal. As an expectation set, this might be expressed by saying that the occupancy frequency on all other states (not on this path) must be zero (or at least close). An example of a human-sufficient reward function would be a reward of +1 for reaching the goal and some negative reward for each time step.
>
> Now assume that, in fact, there is lava in one of the states on the direct path (i.e. this is the correct robot model). If the human knew this, they'd like the robot to avoid the lava. Intuitively, it seems clear that the reward function above is misspecified (since it doesn't include the negative reward for lava). Similarly, the expectation set feels "misspecified" to me, since the only policy that satisfies the expectation set (under the true robot model) walks through lava. So my sense is that both reward functions and expectation sets suffer from exactly the same type of problem here. This is an example of what I had in mind when expressing concerns about whether expectation sets really improve the situation.
>
> My current understanding is that this is *not* a type of misspecification/limited human knowledge you are addressing. Instead, you are *assuming* that the expectation set correctly expresses human preferences, and the only issue arises from incorrect beliefs about the transition function. Is that a good summary?
>
> **Theorem 1/Examples:** I've looked in more detail at Theorem 1, since it seems like it could provide good examples motivating expectation sets from my perspective. (Apologies for not noticing this earlier.) Right now though, I don't follow the proof sketch.
>
> If we only had two expectation elements, <S, >, 0> and <S', =, 0>, I think we could just set the reward on S' to something very negative and on S to something much larger. Then, if there is any policy that satisfies the expectation set, all optimal policies should satisfy the expectation set. (Let me know if that doesn't seem right.) So I think there are two ways to prove Theorem 1:
> 1. Construct a case where *no* policy satisfies the expectation set (under the robot domain). This would make the claim technically true but I'd interpret the meaning quite differently: this wouldn't be about reward misspecification, it would just be about humans having unrealistic expectations that can't be met by any policy.
> 2. Construct a more complex expectation set that doesn't have any representation as a reward function for a more "interesting" reason than just being fundamentally unsatisfiable.
>
> If the proof sketch is meant to do 2., then it's not currently clear to me what this expectation set actually looks like (the part about needing to reward some states differently isn't yet obvious to me).
>
> **Questions I still have:**
> * Did I understand correctly that the example I describe is out of scope for the problems you're trying to solve? If not, could you say more about how expectation sets help in this specific example?
> * The example I describe seems like a simplified version of how I think misspecification in the "Puddle" environment works in your experiments. Is that correct? This makes me think it would *not* be out of scope. Are you modeling things differently than I did above?
> * Could you say more about the proof sketch for Theorem 1? (e.g. describe the actual example similar to my gridworld description above, or at least describe the expectation set more if a full construction is too lengthy)
>
> I realize these are a lot of new questions/notes---your responses have helped me clarify my initial concerns into these hopefully more concrete uncertainties.

---

> > ### Author Response · Authors · 2024-08-09
> > **Response**
> >
> > Again, thank you so much for responding to our comments
> >
> > Example:
> > So, in the example you provided, there is a question about whether the expectation was to reach the goal or to follow the exact path. Let's go with the second case and assume the human's expectation was, in fact, to pass through that exact path. In this case, the robot can never achieve it, as with the reward function. Our method can detect the fact that this cannot be achieved and let humans know that the expectation set cannot be achieved. A purely reward-based system would try to optimize for the best reward estimate (or worse yet, the original reward function); this is what an approach like IRD might do. Even though IRD paper uses the cell with lava as an example, depending on the example, it is possible the average reward need not penalize the lava cells enough to cause the agent to avoid it (for example, there might be other cells of unknown features that weren't there in the human model either). The other advantage of expectations again goes back to the point about transferability and when the expectation set can be satisfied in the robot model. The same reward function might not lead to policies that will satisfy the expectations set in both models. However, different reward functions might exist that would satisfy them in each model. However, the reverse is not true. If the expectation set cannot be achieved in the robot model, no reward function can change that.
> >
> > Yes, we are assuming that the expectation set captures the true human preferences.
> >
> >
> > Theorem1 proof:
> > Again, please note that we are fine with cases where no solutions exist. It is important to identify the problem and inform the user.
> >
> > So, there are multiple ways to construct a counter-example that shows the absence of a reward function that translates across domains, but policy exists. The simplest one is when you have a case where you take the form of rewards to be $R: S \times A \rightarrow \mathbb{R}$ (note the reward form has nothing to do with the generality of MDP formulation or any of our methods). Now, let us assume the states to be achieved and avoided are ones where no actions are available (again consistent with the most general MDP construction, referred to as control constraint in optimal control [1]). Now, from the starting state, it has access to two actions. One that takes you to the goal state and the other one that takes you to the state to be avoided deterministically. Now, in the human model, you have to reward one over the other. Let's assume in the robot model, their dynamics are reversed. Now, the previous reward function is no longer able to achieve the expectation set in the robot model.
> >
> > [1] Bertsekas, Dimitri. Dynamic programming and optimal control: Volume I. Vol. 4. Athena scientific, 2012. (http://www.athenasc.com/DP_Slides_2015.pdf page nine for reference)

---

> > > ### Comment · Reviewer_RJbh · 2024-08-09
> > >
> > > Thanks. I agree that being able to flag when an expectation is unsatisfiable is very useful. I mostly still stand by the high-level concerns I expressed in my original review, but I've become convinced on enough specific points that I'm increasing my score.
> > >
> > > A few remaining points (which I don't expect us to converge on in the remaining time, I'm listing these just in case they're useful when updating the paper):
> > >
> > > > Yes, we are assuming that the expectation set captures the true human preferences.
> > >
> > > This seems like a reasonable assumption for research purposes but I certainly don't think it holds universally. And I think there are likely some cases where humans have an easier time expressing a correct reward functions than having correct expectations about specific behavior (many games with simple goals come to mind as an artificial example). So my current view on this paper is that it's introducing an alternative to reward function as a basis for preferences, and that this alternative is likely sometimes more and sometimes less appropriate. (It may well be that expectation sets are *usually* a better choice, I don't have particularly strong views on that.) If you agree with this interpretation, I would suggest making it more prominent in the presentation.
> > >
> > > On the theorem 1 proof: if I understand this construction correctly, it relies crucially on the fact that there are no actions in the final states and so we can't assign a reward of form R(s, a) to them (and thus have to give the reward in an earlier state). This feels pretty artificial, and hopefully, there's an example that doesn't rely on anything like that (e.g. would also work with no-op actions added everywhere). I think if you can find an example that shows theorem 1 while also being a realistic demonstration of the usefulness of expectation sets, that could be very useful for exposition.

---

> > > > ### Author Response · Authors · 2024-08-09
> > > > **Re: Comment**
> > > >
> > > > We really appreciate the reviewer's taking the time to engage with us. It really gave us a chance to understand the points of potential confusion and where clarification is needed.
> > > >
> > > > True representations of the preferences: We will definitely leave a note that the expectation set doesn't necessarily have to always be the true representation of human preference. Our primary motivating principle was that there could be a separation between what the user would want to see and how the actual instructions/objectives are specified, and that could have implications on how the user should be queried. It is better to query in terms of their true intent as opposed to in terms of the queries themselves. While our choices of expectation set were inspired by results from cognitive science, we agree that those results don't constitute a proof. We will make sure to note that, and it is possible that there could be scenarios where people inherently think in terms of rewards. We will make sure to highlight this distinction and make it part of the narrative.
> > > >
> > > > Running Example that ties to the Proof: Thank you for that suggestion. We agree that it would help with the flow of the paper a lot. We will make sure to include a motivating/running example that directly ties to Theorem 1.
> > > >
> > > > Thank you again for your extremely thoughtful and thorough review and feedback. We really appreciate it.

---

### Author Rebuttal · Authors · 2024-08-06

We thank the reviewer for all the comments and feedback. We are extremely happy that the reviewers found our paper well-written, novel, interesting, and useful. We will make sure to incorporate all suggestions, recommendations, and corrections. We have tried to provide specific answers to each reviewer. However, we wanted to take this global response to address some common points brought about by multiple reviewers. Please note that all citations that don’t start with the ‘ext’ prefix refer to citations from the paper.

Experiments: We want to emphasize that we evaluated against two methods in the experiments. The IRD method tries to find a single reward function, and the MMRQ-k tries to query the user to identify potential constraints. It is worth noting that the baseline query method we used failed to solve a single problem in the fairly generous time limit of 30 minutes we set (the worst-case average time for our method was 224.98 secs). Our choice of IRD was motivated by the fact that it is a prototype of a work that tries to make the case for a single true reward function. Even when it tries to find potential alternate reward function hypotheses that account for the behavior, it will fail to generate violation-free behavior as it doesn’t directly query the user about their expectations. At the same time, MMRQ-k results show how poorly the current query methods scale and the utility of leveraging information about the user's mental model.

Moving Beyond Grid-based Domains: The most direct way to move beyond such problems would be to adopt factored or feature-based representations [ext1]. Such representation schemes are typical in reinforcement learning and inverse reinforcement learning works [ext2]. They are also quite popular in AI-safety works (cf. [Zhang et al. 2018],[Saisubramanian et al. 2022], [Mahmud et al. 2023b]). Feature-based representations allow us to provide compact representations of extremely large state spaces. For example, a set of $k$ binary features can encode $2^k$ states. Under this scheme, the goal states and forbidden states would be identified by a set of features, and any state where those features are true is, in fact, considered a goal or forbidden state. Our formulation can be easily extended to a feature-based representation because one could calculate a feature-based occupancy frequency can be calculated by marginalizing across other features. As such, all the constraints and penalties considered in the various LP formulations can still be applied in this case. More interestingly, many aspects of our methods will be simplified by the use of a feature-based representation. For most standard planning problems, the number of features considered is much smaller than the largest state space considered here. This reduces the number of possible queries (because queries are done once per feature) and the number of tests we need to perform to identify potential forbidden state and goal state candidates (again done only once per feature).

[ext1] Brafman, Ronen I., and Carmel Domshlak. "Factored planning: How, when, and when not." AAAI. Vol. 6. 2006.

[ext2] Ng, Andrew Y., and Stuart Russell. "Algorithms for inverse reinforcement learning." Icml. Vol. 1. No. 2. 2000.

The attached PDF includes a table listing the number of constraints per problem instance. The IRD's violations should be compared against this number.

---

### Decision · Program_Chairs · 2024-09-25

**Decision:**

Accept (poster)

**Comment:**

This paper proposes a new framework for inferring human rewards, based on reasoning about sets. The paper provides theoretical guarantees for the proposed approach, and limited evaluation on gridworlds. The reviewers assigned scores of 7/7/5. Reviewers found the paper novel and very well written. Reviewers raised some conceptual questions that were addressed during the rebuttal. The reviewers also raised concerns about limited experimental evaluation, which were not addressed in the rebuttal; during the reviewer discussion Reviewer ahA3 (the reviewer who assigned the paper the lowest score) said that the new “perspective” on preferences would be “valuable to the community,” believing that the “experiments can improve over time.”  Taking this all into account, I recommend that the paper be accepted.